# Analysis of a Special Sulphite-Producing Yeast Starter after Fermentation and during Wine Maturation

**Diána Sárdy Nyitrai [1], Zsuzsanna Varga [1,*], Annamária Sólyom-Leskó [1], Miklós Kállay [1], Szabina Steckl [1], Balázs Nagy [1], Dorottya Kocsis [1] and Eszter Antal [2]**

1   Institute of Viticulture and Enology, Hungarian University of Agricultural and Life Sciences, Ménesi str. 45, 1118 Budapest, Hungary
2   Diagnosticum Zrt., Attila str. 126., 1047 Budapest, Hungary
*   Correspondence: varga.zsuzsanna@uni-mate.hu

**Abstract:** In the present study, we investigated the extent to which specific sulphur dioxide-producing wine yeasts produce $SO_2$ during alcoholic fermentation and whether the $SO_2$ they produce is sufficient to prevent oxidation during wine storage. Fermentation was carried out at 12 °C and 20 °C. After inoculation with yeast, the evolution of free and total sulphurous acid concentrations, acetaldehyde concentrations (11.6–46.9 mg $L^{-1}$) and total polyphenol concentrations (137.4–244.7 mg $L^{-1}$), including leucoanthocyanin and catechin concentrations (leucoanthocyanidin: 8.5–75.1 mg $L^{-1}$; catechin: 70.8–115.4 mg $L^{-1}$), were investigated during the storage of fermented wines. The amount of free sulphur dioxide was measured between 5 and 10 mg $L^{-1}$. Total sulphuric acid ranged from 6 to 22 mg $L^{-1}$, taking into account the results of the three years studied. The aim of our tests was to observe whether the use of sulphur-producing yeasts during the ageing of fermented wines showed any benefit. The results of all three vintages tested showed that neither the 12 °C nor the 20 °C fermented batches showed any advantage in using sulphur dioxide-producing yeasts. Our results show that there is no clear evidence that the yeast produces sufficient sulphur dioxide during fermentation and that there is no clear demonstrable benefit from its use.

**Keywords:** sulfur-dioxide; dry yeast starter; *Saccharomyces cerevisiae*

## 1. Introduction

Yeasts are the most important micro-organisms in the process of converting grape must into wine as they carry out alcoholic fermentation. In most cases, *Saccharomyces cerevisiae* species are responsible for carrying out this process, due to their fermentative capacity and ethanol tolerance [1]. It has long been known that must contains a wide variety of yeasts, mainly composed of *non-Saccharomyces* species; i.e., wild yeasts, such as *Kloeckera, Pichia, Candida, Metschnikowia* and *Kluyveromyces*. Occasionally, species of other genera (*Zygosaccharomyces, Saccharomycodes, Torulaspora, Dekkera, Schizosaccharomyces, Rhodotorula*) may also be present [2–5]. These yeasts may originate from both the microbial communities of grapes and the cellar environment. Many of the non-*Saccharomyces* yeasts, mainly the *Hanseniaspora, Candida, Pichia* and *Kloeckera* species, spontaneously initiate fermentation of up to 4–5% alcohol. Above this level, they rapidly start to die. *Saccharomyces cerevisiae* yeast cell numbers increase as wild yeasts decline, dominating the middle and final stages of the alcoholic fermentation process [3,4]. *Saccharomyces cerevisiae* strains produce wines with high ethanol concentrations and only small amounts of unfermented sugars remain in the medium [6]. An exception to this is the fermentation of wines with very high sugar content, where fermentation stops at the limit of the strains' alcohol tolerance; this leaves a few percent sugar in the wine. Wine yeasts in higher-sugar environments tend to undergo anaerobic metabolism [7]. In this process, known as glycolysis, the yeast cell obtains energy in the form of ATP (adenosine-5′-triphosphate) by breaking down the sugar;

while the cofactor NAD+ (nicotinamide adenine dinucleotide), which is necessary for enzymatic function, is reduced. The pyruvate synthesised during glycolysis is converted to carbon dioxide and alcohol in the last two steps of alcoholic fermentation in an oxygen-free environment [8].

In addition to ethanol, a significant amount of other primary and secondary metabolites are formed, which are important in shaping the taste and aroma of wines. These include glycerol, succinic acid and other organic acids, esters, aldehydes, ketones, higher alcohols and sulphur-containing organic compounds; the types of which are important not only in their species, but also in their quantitative proportions in the development of harmonious flavor and aroma character [9].

Among the inhibitory metabolites produced by yeasts, sulphur dioxide plays a significant role and is only slightly likely to remain in free form in the medium after fermentation [10]. The rate of sulphur dioxide production by wine yeasts is a strain-dependent property. They can be divided into two groups based on sulphur dioxide production: *Saccharomyces cerevisiae* strains with low sulphur dioxide synthesis; and *Saccharomyces bayanus* strains with high sulphur dioxide content, capable of producing up to 300 mg L$^{-1}$ of sulphite. The magnitude of sulphite production can be influenced by a number of factors in addition to yeast strains, such as initial pH, must composition, available nitrogen source, fermentation temperature and sulphate concentration [11]. Sulphur dioxide is formed as an intermediate metabolic product in yeast cells by sulphite reductase enzymes and is transferred from there to musts. For commercially available starter cultures, it is important to test this property during selection [12].

The production of free sulphur dioxide by 10–10 yeast strains from two distributors was investigated by Pezley [13] in red and white wines. The yeast sulphite synthesis varied between strains; however, in all the tests, the white wines produced significantly more sulphite than the red wines. Moreover, he found that the sulphur dioxide produced by the yeasts was not sufficient to protect the wine and that external supplementation was necessary.

Sulphur dioxide-producing yeasts have also been investigated by Wells and Osborne [14], and their effect on lactic acid bacteria was also tested. They found that the yeasts produced mainly acetaldehyde-bound sulphurous acid. The strains with the highest degree of inhibition were those that produced more sulphur dioxide; thus, it was not other toxic compounds that caused the failure of malolactic fermentation. If we are planning biological acid reduction in our wine, the sulphur dioxide-producing properties of the starter cultures must be taken into account. Biological sulphite produced by yeasts may have a chemical–biochemical stabilizing effect; however, it does not provide protection from a microbiological point of view. Thus, adequate sulphur dioxide dosing is still necessary [15].

The use of sulphur dioxide ($SO_2$) is virtually indispensable in wine-making technology. Sulphur dioxide has antiseptic, reducing (antioxidant), flavor and aroma-preserving and color-stabilizing properties that are beneficial from a wine-making point of view. Although some of these functions can be replaced by alternative means, such as ascorbic acid addition or sterile filtration, the taste- and flavor-preserving effect of sulphurous acid, i.e., the binding of free acetaldehyde formed by oxidation, cannot be replaced by any other substance or means known today. In this way, it helps to prevent undesirable spoilage of the wine. In addition to the undoubted benefits of sulphurous acid, its allergenicity must not be overlooked. In sensitive consumers, it can cause headaches, the aggravation of asthma symptoms and coughing. For this reason, one of the recent developments in wine (and food) technology in general has been to reduce or minimize the use of sulphurous acid. One such option is the use of so-called sulphite-producing yeast cultures. Yeasts are capable of producing sulphur dioxide during alcoholic fermentation.

In our experiments, polyphenols were investigated that are largely derived from grapes. They are one of the most important groups of compounds from an oenological point of view. They affect the stability of wine as they are oxidation-prone, highly antioxidant compounds; when oxidized, they can cause browning and other various precipitates. Their



presence also affects the character of the wine. They are also responsible for the color of wines and shape the taste sensation of wines, such as astringent aromas (tannins). They are most important in red wines and have a small health effect. Sulphur dioxide prevents the rapid oxidation of these components [16].

In view of the above, the following objectives were formulated in our experiments: to investigate whether the sulphur content in wines after alcoholic fermentation still requires sulphurization and whether and to what extent the wines still need to be sulphurized in the longer term; the quantitative evolution of the components closely related to sulphurization—total polyphenols, leucoanthocyanins and catechins—was investigated; we also determined the amount of acetaldehyde, a compound that has a major influence on the organoleptic properties of wines.

We also investigated whether the amount of sulphur dioxide formed after alcoholic fermentation is sufficient to ensure the quality of the wine.

## 2. Materials and Methods

During our experiments, storage kinetic studies were carried out in the research laboratory and cellar of the Department of Enology of the MATE Institute of Viticulture and Enology. For fermentation, we used unsulphured 'Olaszrizling' musts from the Badacsony wine region (Hungary) in 3 different years (2017, 2018, 2019). The yeast used for the experiments was a medium sulphur dioxide-producing *Saccharomyces cerevisiae* (SAFCENO$^{TM}$ VR 44 BIO).

The 'Olaszrizling' musts (10–10 L) were inoculated with sulfur-producing yeast. The yeast dosage was 20 g/hl. The fermentation process was carried out at room temperature, 20 °C and 12 °C. The starter culture was prepared according to the manufacturer's recommendation. As a first step, a packet of yeast was soaked in 5 L of 30–38 °C dechlorinated water for 10 min and mixed to suspend the cells. The yeast suspension was then added to 20 L of unsulphurized must and allowed to stand for 20 min. The activated yeast suspension was then added to the must to be fermented and mixed thoroughly to ensure that the yeast was well suspended.

The initial parameters of the musts made from the grapes used for the experiment were as seen in the following (Table 1):

**Table 1.** Baseline data for the musts used for fermentation in the three vintages.

| Parameter | 2017 | 2018 | 2019 |
|---|---|---|---|
| Titratable acidity (g/L) | 11.7 | 11.9 | 11.6 |
| pH | 3.06 | 3.12 | 3.08 |
| Content of sugar (ref%) | 21.44 | 22.01 | 21.22 |
| Assimilable nitrogen level (mg/L) | 320 | 334 | 312 |

After fermentation, the wines were not sulphurized; stored in a single room at 12 °C cellar temperature; and measured monthly for seven months for total and free sulphur content, total polyphenol content, leucoanthocyanin and catechin, and acetaldehyde content. As a control sample, the must was not inoculated and was allowed to undergo spontaneous fermentation. In the fermented samples, the evolution of free and total sulphur dioxide and the polyphenol composition were measured for seven months.

For the analytical tests, routine methods were chosen that are used in everyday laboratory practice. Statistical evaluation of the results obtained was evaluated by ANOVA one-factor analysis of variance. The results were tested at the 95% significance level ($p = 0.05$).

The measurement methods listed below were used for the studies:

- Content of free sulphur dioxide and total sulphur dioxide [17].

Free sulfur dioxide is determined by direct titration with iodine. The combined sulfur dioxide is subsequently determined by iodometric titration after alkaline hydrolysis. When added to the free sulfur dioxide, it gives the total sulfur dioxide;

- Examination of the polyphenol composition of wines;
- Determination of total polyphenol content using the Folin–Ciocalteu reagent, calibrated for gallic acid [18].

All the phenolic compounds contained in wine are oxidized by the Folin–Ciocalteu reagent. This reagent is formed from a mixture of phosphotungstic acid, $H_3PW_{12}O_{40}$, and phosphomolybdic acid, $H_3PMo_{12}O_{40}$; which, after oxidation of the phenols, is reduced to a mixture of blue oxides of tungsten, $W_8O_{23}$, and molybdenum, $Mo_8O_{23}$. The blue coloration produced has a maximum absorption in the region of 750 nm, and is proportional to the total quantity of phenolic compounds originally present;

- Leucoanthocyanin content was determined spectrophotometrically after heating with a 40:60 mixture of hydrochloric acid–butyl alcohol containing ferrous sulphate [19];
- For the determination of the catechol content, the wine was diluted with alcohol, reacted with sulphuric acid vanillin and measured spectrophotometrically at 500 nm [20];
- Determination of acetaldehyde was carried out by enzyme spectrophotometry (megazyme acetaldehyde assay kit from Neogene) [21].

Acetaldehyde is quantitatively oxidized to acetic acid in the presence of aldehyde dehydrogenase (Al-DH) and nicotinamide-adenine dinucleotide (NAD+). The amount of NADH formed in this reaction is stoichiometric with the amount of acetaldehyde. It is the NADH that is measured by the increase in absorbance at 340 nm.

All the reagents were purchased from Merck KGaA (Darmstadt, Germany). The reagents used EDTA (ethylenediaminetetraacetic acid, *di*-sodium salt), sodium hydroxide, sulfuric acid, starch, iodine solution, Folin–Ciocalteu reagent, anhydrous sodium carbonate, iron (II) sulfate hydrate, hydrochloric acid, 1-butanol, vanillin and ethanol.

The samples were analyzed on a Unicam Helios β UV–Vis scanning spectrophotometer.

## 3. Results

After fermentation, free and total sulphur dioxide concentrations were measured in the fermented wines every month for seven months during storage. Three vintages were studied in our experiments. The variation in free sulphur dioxide content during the storage of fermented wines is shown in the following table, comparing the three vintages (Table 2).

The free sulphur dioxide content was practically undetectable during storage; except in the 3rd and 4th months, when a slight increase was observed. During storage, the free sulphur dioxide content in all three vintages showed an increase in the 3rd and 4th months; this was regardless of whether the samples were spontaneously fermenting or those made with the yeast inoculation. Fermentation temperature also had no effect on the change in free sulphur dioxide content during storage. A decrease was observed between the 4th and 7th month in all the samples and all the vintages. Sulphur dioxide concentrations of 5 mg $L^{-1}$ or less were measured in months 5, 6 and 7. Overall, due to the nature of the method, these values being below 5 mg/L are not the actual free $SO_2$ content; from this, we conclude that the starter culture used did not produce notable amounts of $SO_2$. Irrespective of the fact that 5 mg $L^{-1}$/l is the minimum detection limit of the method used for the determination of the concentration of this method for the determination of free $SO_2$, it is the most accurate method for the determination of $SO_2$ in oenological analysis [22].

**Table 2.** Changes in the free sulphur dioxide content (mg L$^{-1}$) during storage in three vintages (2017, 2018, 2019) (LOD of the method: 5 mg L$^{-1}$).

| Day of Measurement | Fermentation Method | Year 2017 | Year 2018 | Year 2019 |
|---|---|---|---|---|
| | | Free SO$_2$ Concentration (mg L$^{-1}$) | | |
| 1 | spontaneous-cold | <5 | 6 | <5 |
| | inoculated-cold | <5 | <5 | <5 |
| | spontaneous-warm | <5 | <5 | <5 |
| | inoculated-warm | <5 | <5 | <5 |
| 2 | spontaneous-cold | <5 | 5 | <5 |
| | inoculated-cold | <5 | <5 | <5 |
| | spontaneous-warm | <5 | 5 | <5 |
| | inoculated-warm | <5 | <5 | <5 |
| 3 | spontaneous-cold | 8 | 9 | 7 |
| | inoculated-cold | <5 | 5 | <5 |
| | spontaneous-warm | 8 | 7 | 6 |
| | inoculated-warm | 8 | 7 | 6 |
| 4 | spontaneous-cold | 8 | 10 | 6 |
| | inoculated-cold | 8 | 7 | 8 |
| | spontaneous-warm | 7 | 5 | 7 |
| | inoculated-warm | 6 | 6 | 6 |
| 5 | spontaneous-cold | 6 | <5 | 5 |
| | inoculated-cold | 6 | 5 | 5 |
| | spontaneous-warm | 5 | <5 | 5 |
| | inoculated-warm | <5 | 5 | <5 |
| 6 | spontaneous-cold | <5 | <5 | 5 |
| | inoculated-cold | <5 | <5 | <5 |
| | spontaneous-warm | <5 | <5 | <5 |
| | inoculated-warm | <5 | <5 | <5 |
| 7 | spontaneous-cold | <5 | <5 | <5 |
| | inoculated-cold | 5 | 5 | <5 |
| | spontaneous-warm | <5 | <5 | <5 |
| | inoculated-warm | <5 | <5 | <5 |

The variation in the total sulphur dioxide content of the fermented wines over the three years is illustrated in Figures 1–3. In this case, too, the variation in total sulphur dioxide was measured monthly for seven months. In the figures, the quantities in blue are spontaneous cold fermented wines; the quantities in red are spontaneous warm fermented wines; the quantities in green are inoculated cold fermented wines; and the quantities in purple are inoculated warm fermented wines.

In the cold fermented samples, the total sulphur dioxide content decreased significantly during the seven months of storage; while the free sulphur dioxide content showed a slight increase. At the end of the 7th month, the free sulphur dioxide content was measured at the same concentration in both spontaneously fermented and inoculated samples. In contrast, the total sulphur dioxide was detected at higher concentrations in the inoculated samples; however, this difference was not statistically significant.

The variation in total sulphur dioxide content in the heat fermented batches showed an opposite trend to that in the spontaneous and inoculated batches. A slight decrease was observed in the spontaneously fermented batches. The trend was similar in all three vintages studied.

The explanation of our results is that the method, i.e., the determination by iodometric titration, is specific in that it detects not only sulphurous acid, but also other reducing compounds such as simple phenols. In the present case, it is suspected that the very low values (below 10 mg L$^{-1}$) are not indicative of sulphurous acid, but of other reducing compounds. Together with this, it is likely that the higher measurement results, especially

for total sulphurous acid, already indicate real sulphurous acid levels. However, it can be stated that even in this ideal case, the sulphurous acid level produced by yeasts may not be sufficient for the safe storage of wines; thus, the sulphurization cannot be omitted in our present experiments.

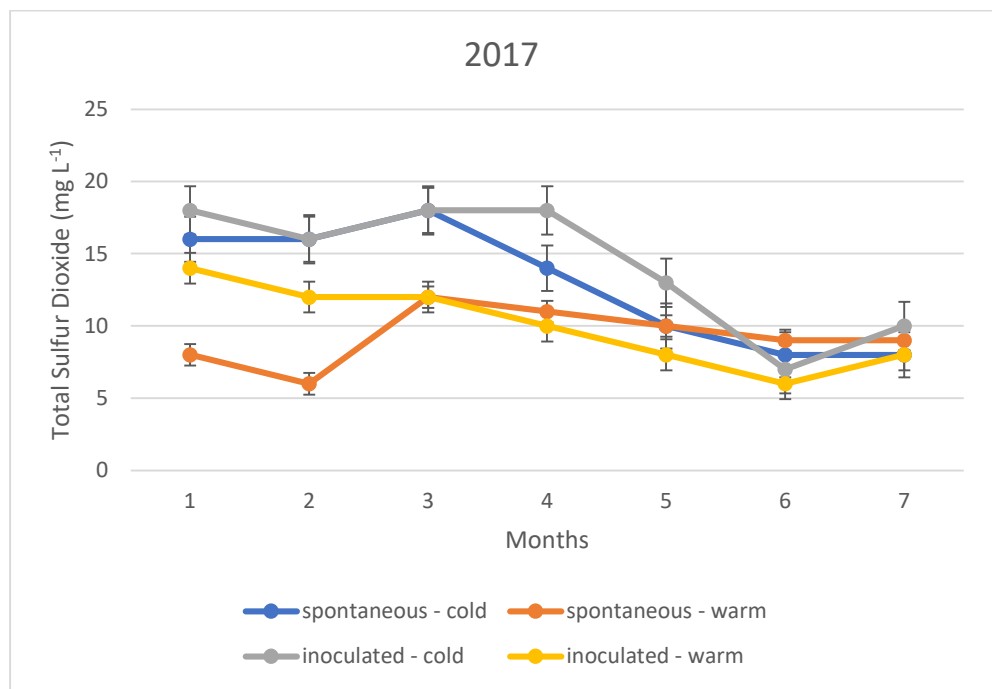

**Figure 1.** Total sulphur dioxide content measured during storage in 2017.

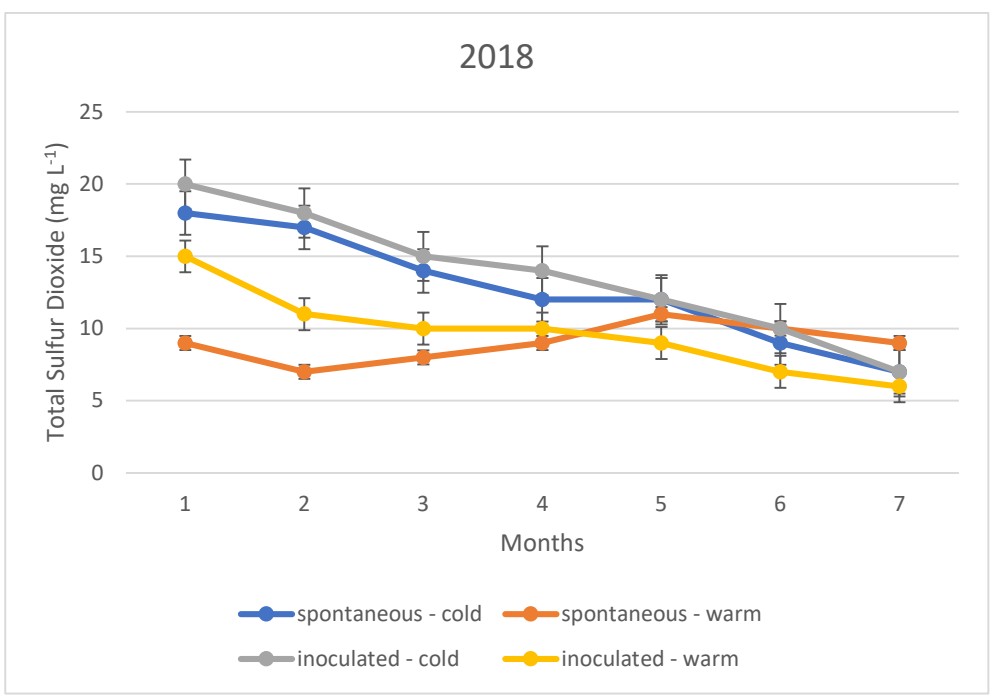

**Figure 2.** Total sulphur dioxide content measured during storage in 2018.

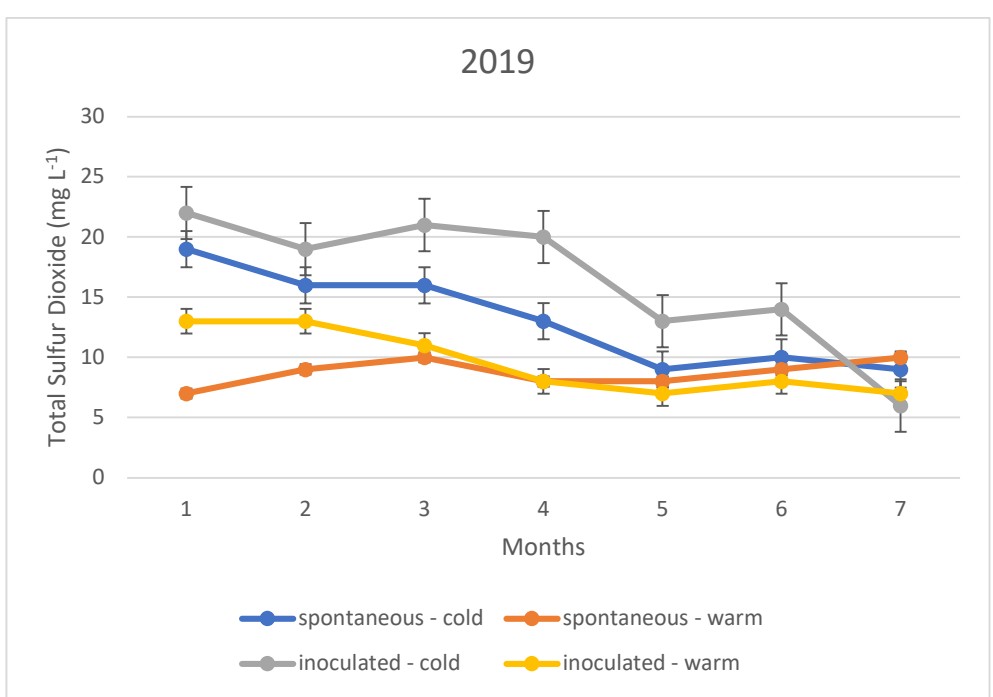

**Figure 3.** Total sulphur dioxide content measured during storage in 2019.

In the three vintages, the total polyphenol content, leucoanthocyanin concentration and catechin content were compared in the fermented wines. In the fermented samples, the aforementioned compounds were compared per vintage according to the degree of fermentation and whether or not the samples were inoculated. Statistical evaluations at the 95% ($p$ = 0.05) significance level clearly show that in all three vintages, the samples inoculated with warm fermented special sulphur-producing yeast had significantly higher total polyphenol content compared to the spontaneously fermented samples. This can be explained by the fact that the amount of sulphur dioxide produced by the sulphur-producing yeast inhibited the oxidation of polyphenols to a significantly greater extent than in the samples fermented at lower temperatures.

The leucoanthocyanin and catechin concentrations in all three vintages were found to be significantly higher in the spontaneously cold fermented samples than in the other samples when tested at the 95% ($p$ = 0.05) significance level. This is the reason why the inoculation with specific yeast is recommended in contrast to spontaneous fermentation; since the amount of bitter flavoring substances and the bitter taste sensation is significantly higher in the spontaneously fermented samples. Based on the polyphenol composition of the wines, no significant difference in the total polyphenol content of the spontaneous samples fermented at different temperatures was found. However, the inoculated samples showed an increase in polyphenol content when fermented at room temperature compared to cold fermentation. The highest values for leucoanthocyanin and catechin were obtained in the cold spontaneously fermented sample. For the inoculated samples, temperature had no significant effect on leucoanthocyanin content; while for catechin, a small increase was observed in the wine fermented at room temperature (Table 3).

The total polyphenol content of wines fermented with a special sulphate starter culture was higher for both fermentation temperatures compared to the spontaneously fermented wines.

The evolution of the leucoanthocyanin and catechin content showed that both components were present in higher concentrations during storage of the cold fermented wines due to the effect of the special starter culture. However, during the maturation of the batches fermented at higher temperatures, the opposite trend was observed: in this case, the leucoanthocyanin and catechin content of the spontaneously fermented samples was higher during maturation.

**Table 3.** Changes in the concentration of phenolic compounds at the end of fermentation in the wines analysed over 3 years (mean ± standard deviation of three parallel measurements).

| Year | Sample Name | Total Polyphenol (mg L$^{-1}$) | Leucoanthocyanidin (mg L$^{-1}$) | Catechin (mg L$^{-1}$) |
|------|-------------|-------------------------------|----------------------------------|------------------------|
| 2017 | spontaneous-cold | 178.2 ± 5.28 | 72.2 ± 2.65 | 109.2 ± 2.86 |
| | spontaneous-warm | 179.2 ± 1.30 | 10.1 ± 1.95 | 73.7 ± 2.52 |
| | inoculated-cold | 167.6 ± 3.82 | 21.9 ± 2.77 | 74.4 ± 3.10 |
| | inoculated-warm | 236.1 ± 7.56 | 18.4 ± 1.61 | 97.6 ± 1.77 |
| 2018 | spontaneous-cold | 145.5 ± 4.04 | 69.6 ± 4.76 | 111.9 ± 2.02 |
| | spontaneous-warm | 147.5 ± 3.91 | 10.8 ± 2.17 | 80.2 ± 2.78 |
| | inoculated-cold | 179.2 ± 3.76 | 21.5 ± 4.33 | 78.2 ± 1.57 |
| | inoculated-warm | 199.2 ± 2.21 | 17.7 ± 1.83 | 89.6 ± 3.94 |
| 2019 | spontaneous-cold | 150.6 ± 3.39 | 70.8 ± 2.46 | 112.1 ± 2.93 |
| | spontaneous-warm | 143.4 ± 5.26 | 15.3 ± 5.38 | 76.3 ± 1.97 |
| | inoculated-cold | 156.9 ± 7.42 | 21.1 ± 3.58 | 75.6 ± 1.95 |
| | inoculated-warm | 204.5 ± 3.30 | 19.4 ± 3.30 | 92.8 ± 2.52 |

The acetaldehyde content during storage showed a different upward trend in all three vintages, regardless of the fermentation temperature and inoculation. This means that the sulphurous acid formed during the experiment is far from sufficient to bind the acetaldehyde produced. The trend suggests that the specific sulphurous acid-producing yeast is no more protective against acetaldehyde formation than the spontaneous yeast activity.

The acetaldehyde content of the wines did not yet reach the organoleptic threshold of 100–125 mg L$^{-1}$ during the seven months of storage; however, an upward trend in concentration is observed for both the cold and warm fermented wines (Figures 4–6). For this reason, rapid sulphurization of the wines is recommended.

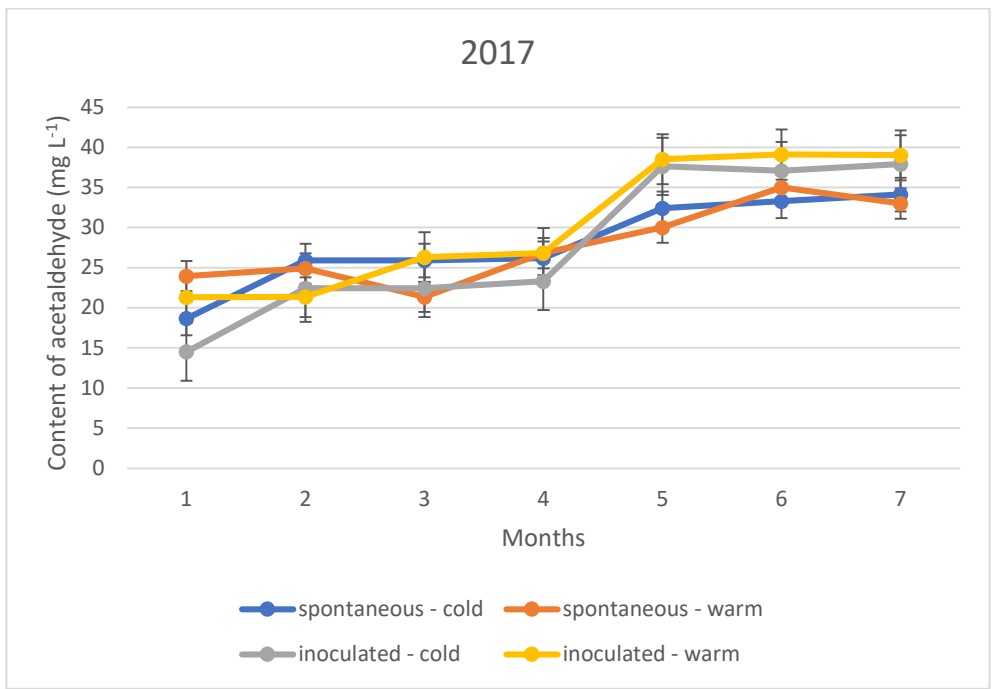

**Figure 4.** Acetaldehyde content of the wines tested during storage in 2017.

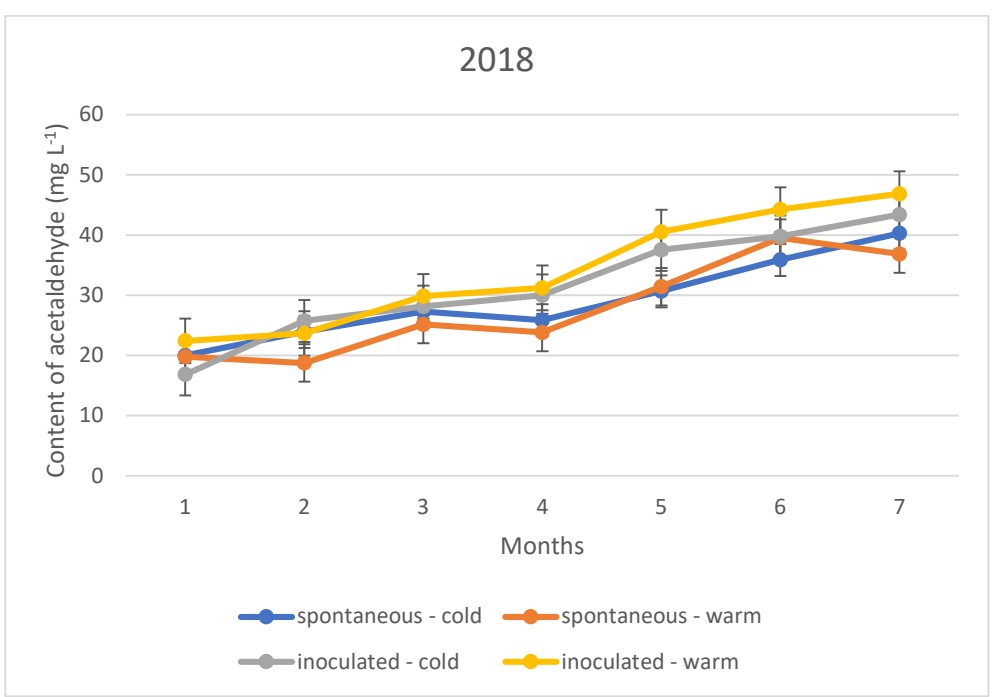

**Figure 5.** Acetaldehyde content of the wines tested during storage in 2018.

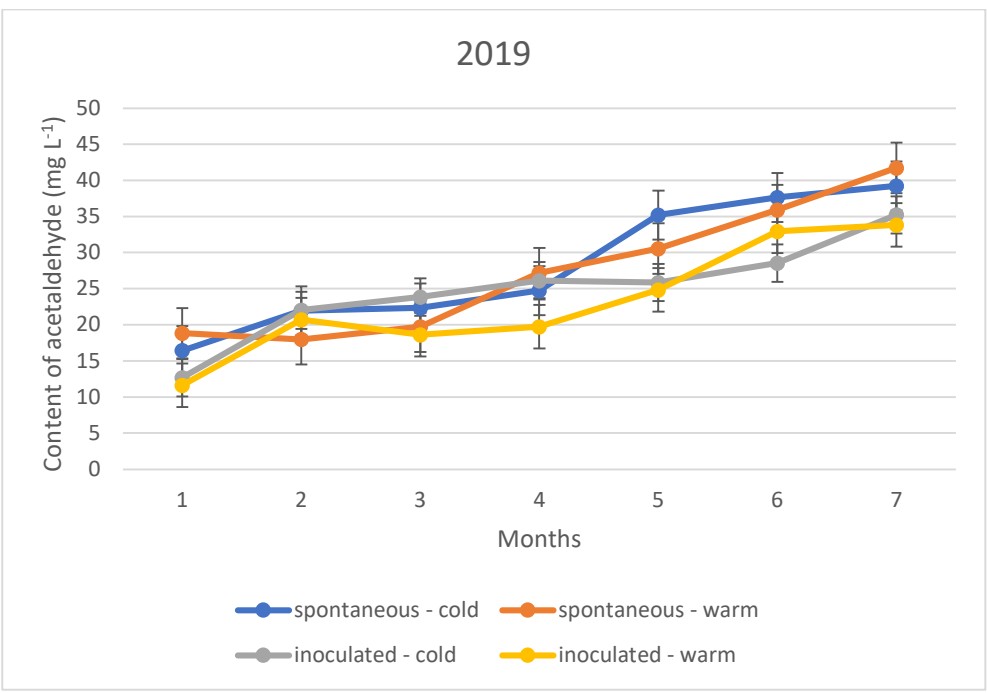

**Figure 6.** Acetaldehyde content of the wines tested during storage in 2019.

## 4. Discussion

Sulphur dioxide and its derivatives are the most widely used auxiliaries in wine-making technology. Originally used for its antimicrobial properties, sulphur dioxide has been used for its antioxidant, aroma, flavor and color-protecting properties. Among the latter, its aroma-preserving and aroma-forming properties should be highlighted, as well as its acetaldehyde-binding properties.

In general, one of the general objectives of current oenological technology is to reduce the sulphur dioxide concentration or even to achieve sulphur-free technology. To this end, modern wine-making technology is now able to filter out micro-organisms, for example, by

using membrane technology; and to control the amount of polyphenolic compounds that can be used as a substrate for oxidation phenomena by using modern (pneumatic) presses in grape processing technology.

However, it is clear from the phenomena discussed above that maintaining a certain concentration of sulphur dioxide, specific to each wine, is essential for the organoleptic properties of wine; this is because it is sulphur dioxide that prevents the rapid oxidation of highly antioxidant components such as polyphenols, in order to avoid, for example, the madeirization and disintegration of white wines. The main reason for the use of sulphur dioxide is therefore to prevent the fragmentation of the wine by absorbing acetaldehyde, which is produced after fermentation and during ageing. This was the subject of our experiments and the results obtained.

Our measurements showed that the use of the so-called sulphurous acid-producing yeast strain we tested did not increase the total sulphurous acid concentration of the wines significantly compared to spontaneous fermentation; however, no similar effect was observed for the free sulphurous acid content in either the cold or the warm fermented batches. During storage, the total sulphurous acid content showed a clear decrease; this confirms previous test results [23]. In an earlier study, Pezley [13] also concluded that the sulphur dioxide produced by yeasts is not sufficient to protect the wine and that external supplementation is needed.

The total polyphenol content is clearly altered by oxidation. The total polyphenol content decreased, since the unoxidized phenolic OH groups are measured by this method; this makes it clear that polyphenols are more oxidized than other previous authors [24].

The acetaldehyde content is also increased due to the extremely low sulphurous acid content. As found by other authors [25], it is clear from our results that the amount of $SO_2$ produced by the yeast we studied is not sufficient to prevent the oxidation and oxygen uptake of wine.

### 5. Conclusions

The use of the sulphur dioxide-producing yeast strains that were tested did not significantly increase the total sulphur dioxide concentration of the wines compared to spontaneous fermentation. Our results suggest that the use of a sulphur dioxide-producing strain does not provide a significant advantage during fermentation. The amount of sulphur dioxide produced will not provide adequate protection from a microbiological point of view. It is still necessary to add the right amount of sulphur dioxide to preserve the quality of the wines.

**Author Contributions:** Conceptualization, D.S.N. and E.A.; methodology, M.K. and B.N.; validation, D.S.N. and E.A.; formal analysis, S.S.; investigation, D.S.N.; resources, D.K.; data curation, D.K.; writing—original draft preparation, E.A.; writing—review and editing, Z.V.; visualization, S.S.; supervision, D.S.N.; project administration, B.N. and A.S.-L.; funding acquisition, D.S.N. All authors have read and agreed to the published version of the manuscript.

**Funding:** This research received no external funding.

**Institutional Review Board Statement:** Not applicable.

**Informed Consent Statement:** Not applicable.

**Data Availability Statement:** The details of the data are available on reasonable request from the corresponding author.

**Conflicts of Interest:** The authors declare no conflict of interest.

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
