# Peer review of "Analysis of a Special Sulphite-Producing Yeast Starter after Fermentation and during Wine Maturation"

_applsci, doi:10.3390/app12178848_

Round 1

Reviewer 1 Report (New Reviewer)

This manuscript lacks a great deal of detail or has other issues (yeast strain used, grape quality and fermentation parameters, inadequate analytical methods, etc.) to make reasonable inferences regarding the data collected.

The quality of English was marginal, with a number of awkward word choices or phrasing. Many paragraphs have far too much information present leading to difficult organization of ideas. It is recommended to have additional review of the order of idea presentation.

Due to these concerns, it is recommended that this manuscript should not be published in its current form. With a large amount of modification, it is possible that portions could be acceptable for the Journal but the authors really must think through their science and the potential inferences based on their data. Specific comments are as follows:

Line                  Comment

13-20               Unclear sentence. How can “the functioning” be investigated? Are the authors investigating “how” the yeast functioned under vinification conditions? What is a “specific sulphite starter culture”? Unsure if “sulphuric acid” is the same as “SO2” (sulfur dioxide) or if the authors are simply expressing the value obtained as sulphuric acid. Third sentence could be moved earlier in abstract. Why weren’t data included in the abstract?

25-85               VERY lengthy paragraph that has too many ideas present. Need to better organize the information. Regarding line 42, is the discussion about Crabtree effect? Line 56 = but vast majority of any SO2 present is probably going to be bound? Unclear rationale presented in line 77-82.

85-97               What is “EC filtration”? Unclear idea expressed in lines 96-97. Is there a difference between sulfites added or those produced by yeasts with regards to consumers? Does a specific concentration of sulfites may or may not cause consumer problems? Seems like this would be a range in concentrations. What about making a wine with a yeast that does not produce SO2 and then just add equivalent amounts post-fermentation? Again, too many ideas in a single paragraph.

98-105              A jump in ideas from previous paragraph?

107-109            Unclear objective. It seems as though the argument should be made as to exactly how much SO2 is needed by a wine during aging, be the sulfites being produced by the yeast or added during the aging period? Somehow, the objectives of the research need to be better clarified…What is different about adding a sulfite-producing yeast as opposed to adding SO2 during aging? All the SO2 concentrations seem low to me.

120                  More information about the yeast is needed (strain number, species, prior publication, etc.)

116-128            Suggest to not begin a sentence with a number. Level of inoculation? Preparation of starter cultures? Why use spontaneous fermentation? There are strains of S. cerevisiae known to not produce much sulfite during fermentation which could serve as better controls. Experimental design including number of replicate fermentations? Degree of ripeness of the grapes (Sugar? Titratable acidity and pH? Yeast assimilable nitrogen level?). Did the fermentations reach dryness? Unclear how “cold” or “warm” fermentations were defined (refer to Figures 1 to 3). What are the dotted lines on each of these figures? Much sulfur can go to H2S as well but that was not measured (did the fermentations smell of rotten eggs?).

135                  Use of the iodine titration is subject to a number of interferences. Does the method have an acceptable detection limit given the intended results? Unsure if the method is reliable enough to measure 1 mg/L accurately given other interferences possible.

184-189            Were these wines in bottles by this time? Any racking during the seven months where SO2 was measured? SO2 will strongly react with oxygen which will be available during racking. Also, was the “intense increase” in the second, third, and fourth months significant? What does the word “intense” imply?

190                  The amounts of total SO2 (20 to 30 mg/L) are low for wines but pH of the ferments is not known either (so molecular SO2 cannot be calculated).

207-210            Unclear.

211-216            If true, then Figures 1 to 3 present questionable data (all concentrations are below 10 mg/L). What are “real” levels as opposed to fake levels?

244                  Data analysis of values (significance?)?

264                  Are the differences in apparent acetaldehyde significant?

275-301            Discussion sections should focus on discussing the results, not necessarily provide additional general information. No references were cited in this section so limited discussion was presented. What do the results actually mean?

303-309            Use of word “significant” was unclear (usually implies statistical analyses). There is no way to know whether the amount of SO2 present was or was not enough for microbial protection (cannot calculate molecular SO2). Only 22 references seems a little low given the amount of work which must have been conducted over the years.

Author Response

Reviewer 2 Report (Previous Reviewer 2)

Authors have made necessary corrections

Author Response

Thank you for your valuable oppinion.

Reviewer 3 Report (New Reviewer)

In the manuscript entitled “Investigation of a specific sulfite-producing yeast starter during fermentation and wine maturation”, the authors investigated the functioning of sulphur-producing yeasts. Both low and high temperature fermented yeasts were studied, and the author found an important result that sulphur dioxide was not sufficiently produced during fermentation. The finding is solid in my opinion.

Author Response

Thank you for your valuable oppinion.

Round 2

Reviewer 1 Report (New Reviewer)

Some of the original Reviewer comments were incorporated; others were not addressed. It is strongly suggested that the Authors be sure to respond to all comments by Reviewers in order to improve the quality of their manuscript.

Abstract: Needs relevant data added or else the reader cannot arrive at the same conclusions as the Authors. Some of the wording in the abstract cold be removed to include at least some data. Last sentence unclear.

Lines 25-85: Very lengthy paragraph with just too many ideas. Needs to be broken up into smaller paragraphs (maybe 2 or 3).

118-119: Without any sensory analyses, can this statement be made?

Line 125: If the supplier cannot provide this information about the yeast, then the manuscript probably should not be published. Research is published with the idea that it can be replicated by other scientists and without basic information about this yeast, how can that happen?

Lines 128-139: Suggest to just write in paragraph form (and can be abbreviated).

Line 160: But does the iodine method yield acceptable detection limits? As per lines 217-221, how can the Authors graph results <5 mg/L on the figures given the limitations of this analytical method? If numbers below 5 mg/L are "not real", then data presented in these figures were also not real.

All figures: Unclear how the dotted lines were developed. These are not necessarily trends given data show different trends.

Lines 217-221: The word "significant" always implies statistical analyses. Is that what is meant here? Paragraph somewhat unclear.

Author Response

This manuscript is a resubmission of an earlier submission. The following is a list of the peer review reports and author responses from that submission.

Round 1

Reviewer 1 Report

The research examined the operation of sulfur-producing yeasts. The research explored the utilization of distinct sulphite starter cultures for wines fermented at low and high temperatures. The development of free and ‘total sulphuric acid concentrations, acetaldehyde concentration, and total polyphenol concentration, including leucoanthocyanin and catechol concentrations were studied after yeast inoculation.

Methods

The name of the sulphur-producing yeast is not indicated

Results

Figure 1: The authors do not explain what 2.2.17, 3.2.17, 6.2.17, 7.2.17, 8.2.17, 16.2.17 and 16.3.17 mean. The sampling points are missing or not explained. This makes the discussion hard to follow

Line165: Not displayed are the findings indicating that the greatest concentration of more than 40 mg L-1 was recorded on the second day of fermentation at 12°C, followed by a gradual decrease. Also not displayed are the sampling points (days).

Figure 4 and 5: The writers are required to address the question of statistical significance between the time points. There do not seem to be any substantial changes in reductivity. No substantial decline in redox status, as stated by the authors.

Figure 7: The actual reductones measured are not specified. The statistical significance is also questionable

Table 1: How statistically significant were these variations

There is a lot of work already been done on sulfite-producing yeast. Several studies (Pezley, 2015; Wells and Osborne, 2011; etc.) have been published on yeast's generation of sulphur dioxide. It is difficult to comprehend the originality of the present work. Even further, other researchers (not referenced in this article) examined the effect of SO2 on wines in relation to wine type and age at varied temperatures.

In addition, the data regarding maturation (as stated by the title) are not presented nor analyzed in detail in the current research.

Author Response

Thank you for your helpful suggestions. Please see the attachment.

Reviewer 2 Report

Authors have conducted investigation of a specific sulfite-producing yeast starter during fermentation and wine maturation. The following corrections to be incorporated. 

Authors need to provide full form of EC in the introduction

Authors need to cite the appropriate reference for the statement “Acetaldehyde is the most abundant aldehyde in wine. It often accounts for more than 90% of the aldehyde content of wine”

Authors need to mention the basis for selecting the fermentation temperature as 12oC and room temperature.

Authors have reported the variation of concentrations of free sulphur dioxide in spontaneous sample and the sample inoculated with yeast for cold and room temperature fermentation. The observations must be supported by valid reasons and must be compared with previously reported data (Figures 1 (a) & (b), Figures 2 (a) & (b), Figures 3 (a) & (b)).

Throughout the manuscript authors have reported the observations made during the experimentation/fermentation. The observations must be supported by valid reasons and the study must report the comparison of the results with the previously reported results.

Table 1 shows the Total Polyphenol, Leucoanthocyanidin, and Catechin concentrations after fermentation at room temperature and cold fermentation. Authors have reported their observations but the proper inference / justification is missing.

Author Response

(The authors gave the same response as above.)

Reviewer 3 Report

The manuscript entitled: “Investigation of a specific sulfite-producing yeast starter during fermentation and wine maturation” reports a study on the functioning of sulphur-producing yeasts in wine making.

The Abstract is short poorly written and should contain also some information in detail of the determined experimental data and on the end points of the manuscript.

TIn general the manuscript is poorly written and some comments should be substantiated, like “the use of sulphur dioxide (SO2) is virtually indispensable in wine-making”: nowadays wines are produce with limited or no sulfur dioxide. The sulfuric acid is used in wine making? See line 57 to 62 and please comment on this point.

The abbreviations should be detailed in full at their first use, se e.g. “E.C.” at line 28, etc. The Introduction seems too long and could be reduced mentioneing the main issues and limits of the evaluated process and go to the manuscript end points and limits directly. This is not clear.

The Materials and Methods section is not fully clear: the Authors mention “fermented wines”: please rephrase. Please add information and details on the experiments performed. Which yeasts have been used and how, in which amount, etc.? Please correlate this information to the Tables reporting the result observed and justify the choices.

What does it mean “higher temperatures” at line 209. Please explain and comment with reference to the reported Figures. Same consideration holds for “cold”: please add information and explain.

Statistical analysis is mentioned but not desribed. The esd (see e.g. Table 1) is missing as well as the number of replication of the analyses performed.

The determination of polyphenols has been mentioned, nonetheless there is question: the develop of alcohol has not been determined? Please explain.

Figure 11 is not clear: please explain better ind etail. The Temperature used is mentioned as “cold” and “warm”: please comment on this.

The Concusion section should avoid repetitions and report the end points and limits of the proposed research. The contribution toarea of interest should be exploited as well as the possible impact on the area of interest.

Too dated References should be avoide unless necessary and justified (e.g. see the one dated 1969, etc.) un favor of more recent literature data available where possible. The mentioned links have been accessed much time ago, it is suggested to check for updated ones when available.

I am afraid to say that in its present form the manuscript cannot be recommended for publication in the Journal.

Author Response

(The authors gave the same response as above.)

Round 2

Reviewer 1 Report

The presentation of the results is still confusing and hard to follow.

Figure1: It remains unknown what the "numbers" 2.2.17, 3.2.17, 6.2.17, 8.2.17, 16.2.17, and 16.3.17 represent.

Line 205: The authors reported a drop of total sulphuric acid at day 15. However, the figure (1) does not show where day 15 is located. The interpretation is still confusing. Additionally, the authors don’t explain the differences between 16.2.17 and 16.3.17 sampling points. This numbering system does not seem to correlate to the number of days as depicted.   

Figure 6: Why the pH reflected under spontaneous cold does not equate to the pH reflected under inoculated cold is still unexplained. There is no thorough explanation of this data in the manuscript.

Figure 7: Why the pH settings in figure 7a do not equate to those in figure 7b is unclear. Moreover, there is no extensive discussion.

The statistics in Table 1 remain unaddressed, hence the findings about the concentration changes cannot be substantiated.

Section 4: The discussion section is generally not supported by the findings.

Reviewer 3 Report

Notwhitstanding the revision made by the Authors, no substantial modification are clearly visible or evidenced in the revised manuscript (see e.g. the yeast used in the Materials and Methods, the origin of the used must, the details regarding experimental procedures, the statistical analysis, “significant difference”, etc.). The end points, contribution to the area of interest and limits of the proposed study should be better detailed and exploited. Moreover, too dated References should be avoided (e.g. see the one dated 1985, etc.) in favor or ore recent literature data unless necessary or substantiated for their use by comments in the text manuscript I am afraid to say that the manuscript cannot be recommended for publication in the Journal.